# HIERARCHY-BASED CLIFFORD GROUP EQUIVARIANT MESSAGE PASSING NEURAL NETWORKS

**Takashi Maruyama & Francesco Alesiani** *
NEC Laboratories Europe
Kurfuerstenanlage 36,
D-69115 Heidelberg, Germany
{Takashi.Maruyama,Francesco.Alesiani}@neclab.eu

## ABSTRACT

We introduce Hierarchy-based Clifford Group Equivariant Message Passing Neural Network (HCGE-MPNN), a Clifford group equivariant U-Net with skip-connection. Our method integrates the expressivity of Clifford group-equivariant layers with hierarchical pooling/unpooling in an encoder-decoder fashion. Our architecture admits major classes of pooling methods, sparse and dense pooling methods. We also introduce a Clifford group invariant projection operator, a generalized projection operator defined on the Clifford space, to make our end-to-end architecture equivariant to Clifford group action. Our method outperforms state-of-the-art (Clifford-)Equivariant MPNNs by up to 7% in prediction MSE for Multi-Nbody datasets and 22% for motion capture dataset.

## 1 INTRODUCTION

Simulation is of huge importance to model and understand multi-body physical systems, such as inorganic material or biological systems, and is widely used to compute properties of the physical systems. However simulation is very sensitive to the accuracy of the description of the governing laws, which typically includes geometric interplay between objects such as symmetry of the systems.

Recent geometric deep learning (Bronstein et al., 2021) has made promising progress for modelling multi-body physical systems. Geometric graphs are typically embedded in a topological space such as a metric space or a manifold. The node features of these graphs include geometric features, like positions or velocities and these quantities are transformed under spatial operations like rotations, reflections, or translations (Cohen & Welling, 2016). E(n)-Equivariant Graph Neural Networks (EG-NNs) (Satorras et al., 2021) are designed to preserve these transformation by imposing an invariance and/or equivariance structure tailored to tasks.

More recently Clifford neural network models (Brandstetter et al., 2022; Melnyk et al., 2021; Ruhe et al., 2023; Spellings, 2022) have been proposed to model high-order interactions of physical systems. Clifford algebra can represent higher-order elements such as bivectors and trivectors and is capable of modeling not only low-order information such as magnitude of vector but high-order information such as area or volume spanned by multiple vectors. The expressive power of the networks with Clifford representation is shown in diverse applications. However, while the Clifford equivariant neural network is capable of higher-order information processing, the networks only conduct *flat* information propagation, which hinders propagation of the spatial and dynamical information.

In this paper, we propose Hierarchical Clifford Group Equivariant Message Passing Neural Network (HCGE-MPNN), an end-to-end trainable model to discover important substructures, respecting Clifford group action (see Section 3). HCGE-MPNN comprises an encoder and decoder equivariant to Clifford group, specifically in our case $E(3)$ group actions (that represents translation and rotations) represented as elements in the algebra. The encoder encodes the input of thee system from fine-scale to coarse-scale and the decoder restore both the topological and numeric information of the input systems. We introduce two *pooling* operations, sparse and dense pooling, and show their equivariance by introducing a novel projection operator on the Clifford space. We demonstrate our

---

*www.neclab.eu.

HCGE-MPNN in two benchmarks of multi n-body systems and motion capture dataset. We show up to 22% reduction in prediction accuracy compared to state-of-the-art geometric group equivariant methods.

## 2 BACKGROUND AND RELATED WORK

**Clifford Algebra and Clifford Neural Networks**. We start by introducing Clifford algebra (Lundholm & Svensson, 2009), also known as geometric algebra (Hestenes & Sobczyk, 2012), over the Euclidean space $\mathbb{R}^n$ and some its properties. We follow similar notation as in Ruhe et al. (2023). The Clifford algebra $Cl(\mathbb{R}^n, q)$ is the quotient space over the tensor algebra over $\mathbb{R}^n$ with an equivalence relation $q(\boldsymbol{v}) = \boldsymbol{v} \otimes \boldsymbol{v}$ ($\forall \boldsymbol{v} \in \mathbb{R}^n$) defined by a quadratic form $q : \mathbb{R}^n \to \mathbb{R}$. Since the anti-commutative relation $\boldsymbol{v} \otimes \boldsymbol{w} + \boldsymbol{w} \otimes \boldsymbol{v} = q(\boldsymbol{v} + \boldsymbol{w}) - q(\boldsymbol{v}) - q(\boldsymbol{w})$ holds for $\forall \boldsymbol{v}, \boldsymbol{w} \in \mathbb{R}^n$, $Cl(\mathbb{R}^n, q)$ is a finite (up to $2^n$) dimensional linear space and every element $\boldsymbol{x} \in Cl(\mathbb{R}^n, q)$ may be written with a finite index $I$ as $\boldsymbol{x} = \sum_{i \in I} c_i \cdot \boldsymbol{v}_{i,1} \otimes \cdots \otimes \boldsymbol{v}_{i,k_i}$, $c_i \in \mathbb{R}$, $\boldsymbol{v}_{i,j} \in \mathbb{R}^n$. Here, the expression $\boldsymbol{v} \otimes \boldsymbol{w}$ of elements $\boldsymbol{v}, \boldsymbol{w} \in \mathbb{R}^n$ represents the *geometric product* of $\boldsymbol{v}, \boldsymbol{w}$, which is a natural generalization of the canonical inner and outer products on $\mathbb{R}^n$. We note that we hereinafter abuse the notation of the tensor product on the tensor algebra for the geometric product. The geometric product defines vector subspaces $Cl^{(m)}(\mathbb{R}^n, q)$ ($m = 0, 1, \cdots, n$), called *grades*, whose elements are $m$-fold geometric product of $\boldsymbol{v} \in \mathbb{R}^n$. We also write $\boldsymbol{x}^{(m)}$ for $\boldsymbol{x} \in Cl(\mathbb{R}^n, q)$ as the component of $\boldsymbol{x}$ belonging to $Cl^{(m)}(\mathbb{R}^n, q)$. An advantage of the Clifford algebra is the flexibility to allow for algebraic representation of physical systems and manipulation of geometric quantities. This geometric arithmetic is injected as an inductive bias to various machine learning models to solve problems in some applications (Melnyk et al., 2021; Spellings, 2022; Brandstetter et al., 2022) and implementation in PyTorch (Alesiani), JAX (Kahlow, a) and TensorFlow (Kahlow, b) are also available. Although the models incorporate various modes of Clifford algebra, it is yet unclear which kinds of symmetry the models can preserve, which hinders the models from being applied to diverse application domains.

**Equivariant Clifford Neural Networks**. Ruhe et al. (2023) first introduces a concept of equivariance for Clifford neural networks and propose Clifford Group Equivariant (graph) neural networks (CGENNs). The key concepts are the Clifford group and its associated representation into the endomorphisms of $Cl(\mathbb{R}^n, q)$. The Clifford group $\Gamma(\mathbb{R}^n, q)$ is a set of invertible multi-vectors belonging to even or odd grades equipped with the induced geometric product. Then a representation $\rho : \Gamma(\mathbb{R}^n, q) \to \text{End}(Cl(\mathbb{R}^n, q))$, called (adjusted) twisted conjugation, is defined and it induces the following equivariant property: for any multivariate polynomial $F \in \mathbb{R}[T_1, \cdots, T_l]$, we have

$$\rho(\boldsymbol{\omega})F(\boldsymbol{x}_1, \cdots, \boldsymbol{x}_l) = F(\rho(\boldsymbol{\omega})\boldsymbol{x}_1, \cdots, \rho(\boldsymbol{\omega})\boldsymbol{x}_l), \quad \forall \boldsymbol{x}_1, \cdots, \boldsymbol{x}_l \in Cl(\mathbb{R}^n, q), \, \forall \boldsymbol{\omega} \in \Gamma(\mathbb{R}^n, q),$$

in which the geometric product is peformed as product. It is worth noting that when the target of $\rho$ is restricted to the base space $\mathbb{R}^n$, $\rho$ induces an isomorphism between $\Gamma(\mathbb{R}^n, q)$ and the orthogonal group on $\mathbb{R}^n$ up to scalar $\mathbb{R}^\times$. This means $\rho$ defines a high-order action on the Clifford space, while it models the Euclidean rotational action on the underlying Euclidean space. Although CGENNs can respect the symmetry of physical systems while allowing for representation of higher-order interaction of objects, CGENNs (and non-equivariant Clifford neural networks) still only conduct flat information propagation.

## 3 HIERARCHY-BASED CLIFFORD GROUP EQUIVARIANT MESSAGE PASSING NEURAL NETWORKS (HCGE-MPNN)

**Notaion**. Each input representing multi-body system is represented as a graph $G(V, E)$ consisting of $N$ nodes $V$, representing particles, and the set of edges $E \subset V^{\times 2}$, where each edge represents the existence of multi-body interactions. Each node $i$ is assigned with a feature $(\boldsymbol{z}_i^{(0)}, \boldsymbol{h}_i^{(0)})$: $\boldsymbol{z}_i^{(0)}$ represents a directional vector in $\mathbb{R}^{mn}$, such as the concatenation of location and velocity of particles, and $\boldsymbol{h}_i^{(0)} \in \mathbb{R}^s$ is the non-directional feature, such as the charges or category types of the atoms. We embed both features into $Cl(\mathbb{R}^n, q)$, and denote corresponding embeddings by $(\tilde{\boldsymbol{z}}_i^{(0)}, \tilde{\boldsymbol{h}}_i^{(0)}) \in Cl(\mathbb{R}^n, q)^{\times (m+s)}$. Note that, we here assume the natural extension of the group action $\rho(\boldsymbol{\omega})$ to the product space $Cl(\mathbb{R}^n, q)^{\times (m+s)}$ and $\boldsymbol{z}_i^{(0)}$ is equivariant to the orthogonal action while $\boldsymbol{h}_i^{(0)}$ is invariant to the action.

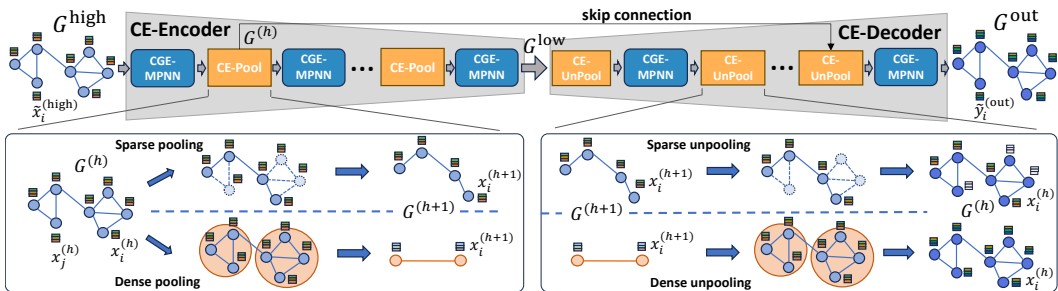

Figure 1: **Schematic diagram of HCGE-MPNN.** HCGE-MPNN is composed of an encoder and a decoder. Each module has corresponding Clifford group equivariant pooing and unpooling layers.

**Problem definition**. Our task is to *predict* the future state of system at a certain time $T$, given directional state $\boldsymbol{z}_i^{(0)}$, that is, to find a model $\Phi$ able to predict a future state $\boldsymbol{z}_i^{(T)}$ for given $T$ and $\boldsymbol{z}_i^{(0)}$. Our model $\Phi$ should be equivariant to Clifford group action, which means

$$\rho(\boldsymbol{\omega})\boldsymbol{z}_i^{(T)} = \Phi(\{\rho(\boldsymbol{\omega})(\tilde{\boldsymbol{z}}_i^{(0)}, \tilde{\boldsymbol{h}}_i^{(0)})\}_{i=1}^N) = \rho(\boldsymbol{\omega})\Phi(\{(\tilde{\boldsymbol{z}}_i^{(0)}, \tilde{\boldsymbol{h}}_i^{(0)})\}_{i=1}^N), \quad \forall \boldsymbol{\omega} \in \Gamma(\mathbb{R}^n, q).$$

When $\Phi$ is based on message passing neural networks, we formulate HCGE-MPNN as an encoder-decoder architecture with skip-connection:

$$\{\boldsymbol{v}_i^{\text{low}}\}, G^{\text{low}} = \text{CE-Encode}(\{\tilde{\boldsymbol{x}}_i^{\text{high}}\}, G^{\text{high}}), \quad \{\boldsymbol{y}_i^{\text{out}}\}, G^{\text{out}} = \text{CE-Decode}(\{\boldsymbol{v}_i^{\text{low}}\}, G^{\text{low}}),$$

which perform the pooling and unpooling operations (see also Figure 1), with $\tilde{\boldsymbol{x}}_i = (\tilde{\boldsymbol{z}}_i, \tilde{\boldsymbol{h}}_i)$, and the encoded and output features $\boldsymbol{v}_i^{\text{low}}, \boldsymbol{y}_i^{\text{out}}$. The encoder applies multiple poolings to nodes in $G$ to capture the multi-scale features of the graph. Specifically, CE-Encode performs pooling operations **CE-Pool**$^{(h)}$ and message passing with Clifford Group Equivariant MPNNs (CGE-MPNNs, Ruhe et al. (2023)) iteratively, up to $H$ times. The decoder maps information from the low level ($h = H$) to the high level graphs ($h = 1$) using the unpooling operator **CE-UnPool**$^{(h)}$.

### CLIFFORD GROUP EQUIVARIANT POOLING AND UNPOOLING

**CE-Pool**$^{(h)}(1 \le h \le H)$ is responsible for making decision on which nodes in a graph are pooled and reconstructing a graph according to the pooled nodes. The pooling process is composed of three processes: *scoring* stage, *node creation* stage, and *topology reconstruction* stage. In the scoring stage, our model first takes a graph $G^{(h)}$ as an input and computes scores $\{r_i\}_{i \in V^{(h)}}$ over all the nodes. In the node creation stage, new nodes of a pooled graph are generated based on the scores computed in the scoring stage. Here, we can take two ways to perform node creation, *sparse* and *dense* pooling methods. After the new nodes are generated, edges are reconstructed. We will mainly discuss the sparse method in the main text, while the dense method is explained in Appendix A.

**Clifford group-invariant scoring function**. Scores $\{r_i\}_{i \in V^{(h)}}$ need to be invariant to Clifford group action since pooled graphs should have same topological structure up to equivariant configurations. To meet this requirement, we introduce *multi-blade projection* over $Cl(\mathbb{R}^n, q)^{\times C}$:

$$f_{\boldsymbol{\theta}}(\boldsymbol{x}) = \sum_{c:\text{channel}} \frac{(\boldsymbol{x}_c \otimes p_{\boldsymbol{\theta}}(\boldsymbol{x}_c))^{(0)}}{\sum \|p_{\boldsymbol{\theta}}^{(0)}(\boldsymbol{x}_c)\|}, \quad \boldsymbol{x} = (\boldsymbol{x}_1, \ldots, \boldsymbol{x}_C) \in Cl(\mathbb{R}^n, q)^{\times C}$$

Here, $p_{\boldsymbol{\theta}}$ denotes a trainable endomorphism of $Cl(\mathbb{R}^n, q)$ and $\|p_{\boldsymbol{\theta}}^{(0)}(\boldsymbol{x})\|$ denotes $\|(p_{\boldsymbol{\theta}}(\boldsymbol{x}) \otimes p_{\boldsymbol{\theta}}(\boldsymbol{x}))^{(0)}\|$. We then have the following theorem for the multi-blade projection:

**Theorem 3.1.** *Let $\boldsymbol{x}, \boldsymbol{p} \in \mathbb{R}^n$ and $C$ a positive integer. Then,*

(i) *if $C = 1$ and $p_{\boldsymbol{\theta}}(\tilde{\boldsymbol{x}}) = \tilde{\boldsymbol{p}}, \boldsymbol{x}_c = \tilde{\boldsymbol{x}} \in Cl^{(1)}(\mathbb{R}^n, q)$, then the multi-blade projection is the projection of $\boldsymbol{x}$ over $\boldsymbol{p}$ in $\mathbb{R}^n$.*

(ii) *If $p_{\boldsymbol{\theta}}$ is equivariant to $\Gamma(\mathbb{R}^n, q)$, the multi-blade projection is invariant to $\Gamma(\mathbb{R}^n, q)$.*

The proof can be found in Appendix C. Theorem 3.1 indicates that the the multi-blade projection is a generalization of the projection defined on $\mathbb{R}^n$ and guaranties to generate same scores no matter how the Clifford group acts on multi-vectors. We will use the multi-blade projection $f_{\boldsymbol{\theta}}$ as the score function in the rest of the paper.

**Sparse pooling and unpooling**. Sparse pooling $\mathbf{CE}\text{-}\mathbf{Pool}^{(h)}_{sparse}$ first computes a score for each node (requiring $O(N)$ space), and reduce a graph $G^{(h)}$ into $G^{(h+1)}$ by keeping only the top $\lceil rN \rceil$ scoring ones and dropping the rest. Here $r$ is the ratio that limits the amount of pooled nodes and given as a hyperparamter. We store all the edges $E^{(h)}$ and $(\tilde{\boldsymbol{z}}_i^{(h)}, \tilde{\boldsymbol{h}}_i^{(h)})$ of dropped nodes. The stored information are used when unpooling ($\mathbf{CE}\text{-}\mathbf{UnPool}^{(h)}$) $G^{(h)}$ to $G^{(h-1)}$: $E^{(h)}$ is used to restore graph topology for $G^{(h-1)}$ and the restored nodes are complemented with some $\Gamma(\mathbb{R}^n, q)$-invariant node features such as mean of node features at $G^{(h)}$ or zero vectors. The overall pooling architecture is similar to TopK-pooling (Gao & Ji, 2019; Cangea et al., 2018), but TopK-pooling is not rotational equivariance since scoring function is not $O(n)$-invariant.

**Dense Pooling and Unpooling**. Dense pooling $\mathbf{CE}\text{-}\mathbf{Pool}^{(h)}_{dense}$ is an extension of the multi-blade projection, in which all nodes in $G^{(h)}$ are explicitly assigned to a pooled node in $G^{(h+1)}$. We first pre-define a number of clusters, which we denote as $K^{(h+1)}$ ($K^{(1)} = N$), as the number of nodes for the pooled graph $G^{(h+1)}$. To ensure the equivariance $\mathbf{CE}\text{-}\mathbf{Pool}^{(h)}_{dense}$, we then compute confidence scores with the multi-blade projection, but here we have the same number of multi-blade projections $(f_{\boldsymbol{\theta}}^{(k)}(\boldsymbol{x}^{(h)}))_{1 \le k \le K^{(h+1)}}$ as the clusters ($= K^{(h+1)}$) and we define the $K^{(h)} \times K^{(h+1)}$ assignment matrix $\boldsymbol{S}^{(h)} = \{S_{i,k}^{(h)}\}$ as follows:

$$S_{i,k}^{(h)} = \left( f_{\boldsymbol{\theta}}^{(k)}(\boldsymbol{x}_i^{(h)}) = \sum_{c:\text{channel}} \frac{(\boldsymbol{x}_{i,c}^{(h)} \otimes p_{\boldsymbol{\theta}^{(k)}}(\boldsymbol{x}_{i,c}^{(h)}))^{(0)}}{\sum_c \|p_{\boldsymbol{\theta}^{(k)}}^{(0)}(\boldsymbol{x}_{i,c}^{(h)})\|} \right)_{1 \le i \le K^{(h)}, 1 \le k \le K^{(h+1)}}.$$

We assign nodes in $G^{(h)}$ to a cluster according to $\boldsymbol{S}^{(h)}$, possibly with channel-wise normalization. Then, the node feature and edges of $G^{(h+1)}$ are computed as follows:

$$\boldsymbol{X}^{(h+1)} = \left( \mathbb{1} \cdot \left( \frac{1}{\sum_{i=1}^{K^{(h)}} S_{i,j}^{(h)}} \right)_j^{\mathrm{T}} \right) \odot (\boldsymbol{S}^{(h)})^{\mathrm{T}} \boldsymbol{X}^{(h)}, \quad \mathbb{1} \in \mathbb{R}^{K^{(h)}} \tag{1}$$

$$\boldsymbol{A}^{(h+1)} = (\boldsymbol{S}^{(h)})^{\mathrm{T}} \boldsymbol{A}^{(h)} \boldsymbol{S}^{(h)}. \tag{2}$$

When performing the dense pooling by (4) and (5), we also store $E^{(h)}$, and use the stored edges' information as edges for $G^{(h+1)}$ to be pooled while restoring node features as follows:

$$\boldsymbol{X}^{(h)} = \boldsymbol{S}^{(h)} \boldsymbol{X}^{(h+1)}, \quad \boldsymbol{X}^{(h)\mathrm{T}} = (\boldsymbol{x}_1^{(h)}, \dots, \boldsymbol{x}_{K^{(h)}}^{(h)}) \tag{3}$$

These operations are similar to the method proposed in Han et al. (2022), but their architecture is Euclidean group-equivariant, while our method is equivariant to the Clifford group action that can respect high-order symmetry of systems.

For these two sparse and dense (un)pooling methods, we can impose equivariance to Clifford-group action, by adopting multi-blade projection. The claim is derived as a corollary of Theorem 3.1, whose proofs are given in Appendix C.

**Corollary 3.2.** *Let $p_{\boldsymbol{\theta}}$ be a trainable $\Gamma(\mathbb{R}^n, q)$-equivariant endomorphism of $Cl(\mathbb{R}^n, q)$. Then, the sparse and dense (un)poolings are $\Gamma(\mathbb{R}^n, q)$-equivariant.*

**Training objective**. We define the training objective of HCGE-MPNN, depending on the pooling modes:

$$L = \begin{cases} \frac{1}{N} \sum_{i=1}^N \|(\boldsymbol{y}_i^{out})^{(1)} - \boldsymbol{y}_i^{gt}\|_p^p, & \text{if sparse} \\ \sum_{i=1}^N \|(\boldsymbol{y}_i^{out})^{(1)} - \boldsymbol{y}_i^{gt}\|_F + \lambda \sum_{h=1}^H \|(\boldsymbol{S}^{(h)})^{\mathrm{T}} \boldsymbol{A}^{(h)} \boldsymbol{S}^{(h)}\|_F. & \text{if dense} \end{cases}$$

Here, the norm for the sparse pooling mode represents $l_p$-norm and we found that $p = 2$ achieves the best performance. We use the Frobenius norm instead for the dense pooling mode. We call the second term (without the weight $\lambda$) in the dense pooling mode *connectivity loss*. The purpose of having this term is to encourage more connections within the pooling nodes and smaller cuts among clusters (Ying et al., 2018).

Table 1: **Prediction error** ($\times 10^{-2}$) **on Multi N-Body dataset**. The "Multiple System" contains $J = 5$ different systems. For each column, $(M, N/M)$ indicates that each system contains $M$ complexes of average size $N/M$. Results averaged across 3 runs.

|  | Single System ($J = 1$) | | Multiple Systems ($J = 5$) | |
|---|---|---|---|---|
|  | (3,3) | (5,5) | (3,3) | (5,5) |
| EGNN (Satorras et al., 2021) | 12.69 (0.19) | 15.37 (0.13) | 13.33 (0.12) | 15.48 (0.16) |
| EGHN (Han et al., 2022) | 11.58 (0.10) | 14.42 (0.80) | 12.80 (0.56) | 14.85 (0.30) |
| CGE-MPNN (Ruhe et al., 2023) | 9.44 (0.41) | **8.18** (0.14) | 9.60 (0.21) | **7.18** (0.36) |
| HCGE-MPNN (sparse) | **8.75** (0.24) | 8.21 (0.13) | **9.14** (0.40) | 7.29 (0.61) |

## 4 EXPERIMENTS

**Multi N-Body**. The Multi N-Body dataset is based on (Kipf et al., 2018; Han et al., 2022) and it consists of $N$ charged particles $\{x_i, v_i, c_i\}_{i=1}^{N}$ that belongs to $M$ disjoint complex systems $\{\mathcal{S}_j\}_{j=1}^{M}$, where $x_i, v_i, c_i$ are particles' position, velocity and change. The task consists of predicting the final position of the all the particles after $T = 1500$ steps, given the initial positions and velocities. The accuracy of the prediction is measure in mean-squared error (MSE). The detail for the experiment is described in Appendix D. The results are reported in Table 1. Our method outperforms the two baselines and also perform better than CGE-MPNN for most of the settings. Even for the larger systems such as (5, 5)-systems, the method is competitive to CGE-MPNN. The results indicate the efficacy of employing hierarchical structure of underlying system together along with arithmetic of the Clifford algebra as inductive bias for the neural network, but also reveals its limitation. The more discussion of the result and the visualization of the results can be found in Appendix E and H

**Motion Capture**.

We also evaluate our model on CMU Motion Capture Database CMU (2003), especially on *walking* (Subject # 35) dataset, in the task of predicting future configuration. For each of the models, we run experiments 3 times and average the results. The detail for the experiment is described in Appendix D. The results are shown in Table 2. We see that HCGE-MPNN outperforms the other strong baselines and the dense mode achieves the best among the other models. The number of parameters for HCGE-MPNN are around 70,000 which is comparable to other baselines. The more discussion of the result and the visualization of the results can be found in Appendix E and H.

Table 2: **Prediction error MSE** ($\times 10^{-2}$) **on motion-capture dataset**. Results are averaged across 3 runs.

|  | Walk (#35) |
|---|---|
| EGNN (Satorras et al., 2021) | 7.9 (0.4) |
| EGHN (Han et al., 2022) | 7.2 (0.1) |
| CGE-MPNN (Ruhe et al., 2023) | 7.7 (0.5) |
| HCGE-MPNN (sparse) | 6.3 (0.2) |
| HCGE-MPNN (dense) | **6.0 (0.1)** |

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

## APPENDIX

## A  CLIFFORD EQUIVARIANT DENSE POOLING AND UNPOOLING

Dense pooling $\mathbf{CE\text{-}Pool}_{dense}^{(h)}$ is an extension of the multi-blade projection, in which all nodes in $G^{(h)}$ are explicitly assigned to a pooled node in $G^{(h+1)}$. We first pre-define a number of clusters, which we denote as $K^{(h+1)}$ ($K^{(1)} = N$), as the number of nodes for the pooled graph $G^{(h+1)}$. To ensure the equivariance $\mathbf{CE\text{-}Pool}_{dense}^{(h)}$, we then compute confidence scores with the multi-blade projection, but here we have the same number of multi-blade projections $(f_{\boldsymbol{\theta}}^{(k)}(\boldsymbol{x}^{(h)}))_{1 \le k \le K^{(h+1)}}$ as the clusters ($= K^{(h+1)}$) and we define the $K^{(h)} \times K^{(h+1)}$ assignment matrix $\boldsymbol{S}^{(h)} = \{S_{i,k}^{(h)}\}$ as follows:

$$S_{i,k}^{(h)} = \left(f_{\boldsymbol{\theta}}^{(k)}(\boldsymbol{x}_i^{(h)}) = \sum_{c:\text{channel}} \frac{(\boldsymbol{x}_{i,c}^{(h)} \otimes p_{\boldsymbol{\theta}^{(k)}}(\boldsymbol{x}_{i,c}^{(h)}))^{(0)}}{\sum_c \|p_{\boldsymbol{\theta}^{(k)}}^{(0)}(\boldsymbol{x}_{i,c}^{(h)})\|}\right)_{1 \le i \le K^{(h)}, 1 \le k \le K^{(h+1)}}.$$

We assign nodes in $G^{(h)}$ to a cluster according to $\boldsymbol{S}^{(h)}$, possibly with channel-wise normalization. Then, the node feature and edges of $G^{(h+1)}$ are computed as follows:

$$\boldsymbol{X}^{(h+1)} = \left(\mathbb{1} \cdot \left(\frac{1}{\sum_{i=1}^{K^{(h)}} S_{i,j}^{(h)}}\right)_j^{\mathrm{T}}\right) \odot (\boldsymbol{S}^{(h)})^{\mathrm{T}} \boldsymbol{X}^{(h)}, \quad \mathbb{1} \in \mathbb{R}^{K^{(h)}} \tag{4}$$

$$\boldsymbol{A}^{(h+1)} = (\boldsymbol{S}^{(h)})^{\mathrm{T}} \boldsymbol{A}^{(h)} \boldsymbol{S}^{(h)}. \tag{5}$$

When performing the dense pooling by (4) and (5), we also store $E^{(h)}$, and use the stored edges' information as edges for $G^{(h+1)}$ to be pooled while restoring node features as follows:

$$\boldsymbol{X}^{(h)} = \boldsymbol{S}^{(h)} \boldsymbol{X}^{(h+1)}, \quad \boldsymbol{X}^{(h)\mathrm{T}} = (\boldsymbol{x}_1^{(h)}, \dots, \boldsymbol{x}_{K^{(h)}}^{(h)}) \tag{6}$$

These operations are similar to the method proposed in Han et al. (2022), but their architecture is Euclidean group-equivariant, while our method is equivariant to the Clifford group action that can respect high-order symmetry of systems.

## B  TRAINING OBJECTIVE

We define the training objective of HCGE-MPNN, depending on the pooling modes:

$$L = \begin{cases} \frac{1}{N} \sum_{i=1}^N \|(\boldsymbol{y}_i^{out})^{(1)} - \boldsymbol{y}_i^{gt}\|_p^p, & \text{if sparse} \\ \sum_{i=1}^N \|(\boldsymbol{y}_i^{out})^{(1)} - \boldsymbol{y}_i^{gt}\|_F + \lambda \sum_{h=1}^H \|(\boldsymbol{S}^{(h)})^{\mathrm{T}} \boldsymbol{A}^{(h)} \boldsymbol{S}^{(h)}\|_F. & \text{if dense} \end{cases}$$

Here, the norm for the sparse pooling mode represents $l_p$-norm and we found that $p = 2$ achieves the best performance. We use the Frobenius norm instead for the dense pooling mode. We call the second term (without the weight $\lambda$) in the dense pooling mode *connectivity loss*. The purpose of having this term is to encourage more connections within the pooling nodes and smaller cuts among clusters (Ying et al., 2018).

## C   PROOF OF EQUIVARIANCE

*Proof of Theorem 3.1.* (i) Since $p_{\boldsymbol{\theta}}(\tilde{\boldsymbol{x}}) = \tilde{\boldsymbol{p}}$ and $\boldsymbol{x}_c = \tilde{\boldsymbol{x}} \in Cl^{(1)}(\mathbb{R}^n, q)$, $\tilde{\boldsymbol{x}}$ and $\tilde{\boldsymbol{p}}$ may be written as

$$\tilde{\boldsymbol{x}} = \sum_{i=1}^{n} x_i \boldsymbol{e}_i, \quad \tilde{\boldsymbol{p}} = \sum_{i=1}^{n} p_i \boldsymbol{e}_i, \quad x_i, p_i \in \mathbb{R}.$$

Here, $\{\boldsymbol{e}_i\}$ is the canonical orthogonal basis of $\mathbb{R}^n$. Then, by taking the geometric product,

$$\tilde{\boldsymbol{x}} \otimes p_{\boldsymbol{\theta}}(\tilde{\boldsymbol{x}}) = \tilde{\boldsymbol{x}} \otimes \tilde{\boldsymbol{p}}$$
$$= (\sum_{i=1}^{n} x_i \boldsymbol{e}_i) \otimes (\sum_{i=1}^{n} p_i \boldsymbol{e}_i)$$
$$= \sum_{i=j}^{n} x_i p_j \boldsymbol{e}_i \otimes \boldsymbol{e}_j + \sum_{i \neq j}^{n} x_i p_j \boldsymbol{e}_i \otimes \boldsymbol{e}_j$$
$$= \sum_{i=1}^{n} x_i p_i + \sum_{i<j}^{n} (x_i p_j - x_j p_i) \boldsymbol{e}_i \otimes \boldsymbol{e}_j,$$

and therefore we get

$$(\tilde{\boldsymbol{x}} \otimes p_{\boldsymbol{\theta}}(\tilde{\boldsymbol{x}}))^{(0)} = \sum_{i=1}^{n} x_i p_i.$$

In a manner similar to the above derivation, we can also deduce

$$(p_{\boldsymbol{\theta}}(\tilde{\boldsymbol{x}}) \otimes p_{\boldsymbol{\theta}}(\tilde{\boldsymbol{x}}))^{(0)} = \sum_{i=1}^{n} p_i p_i.$$

Therefore,

$$\frac{(\tilde{\boldsymbol{x}} \otimes p_{\boldsymbol{\theta}}(\tilde{\boldsymbol{x}}))^{(0)}}{(p_{\boldsymbol{\theta}}(\tilde{\boldsymbol{x}}) \otimes p_{\boldsymbol{\theta}}(\tilde{\boldsymbol{x}}))^{(0)}} = \frac{\sum_{i=1}^{n} x_i p_i}{\sum_{i=1}^{n} p_i p_i} = \frac{<\boldsymbol{x}, \boldsymbol{p}>}{<\boldsymbol{p}, \boldsymbol{p}>}.$$

(ii) Note that in general, for $\forall \boldsymbol{x}, \boldsymbol{y} \in Cl(\mathbb{R}^n, q)$, we have the following relation

$$(\rho(\boldsymbol{w})\boldsymbol{x} \otimes \rho(\boldsymbol{w})\boldsymbol{y}) = \rho(\boldsymbol{w})(\boldsymbol{x} \otimes \boldsymbol{y}), \quad \forall \boldsymbol{w} \in \Gamma(\mathbb{R}^n, q).$$

Then,

$$(\rho(\boldsymbol{w})\boldsymbol{x}) \otimes (p_{\boldsymbol{\theta}}(\rho(\boldsymbol{w})\boldsymbol{x})) = (\rho(\boldsymbol{w})\boldsymbol{x}) \otimes (\rho(\boldsymbol{w})p_{\boldsymbol{\theta}}(\boldsymbol{x}))$$
$$= \rho(\boldsymbol{w})(\boldsymbol{x} \otimes p_{\boldsymbol{\theta}}(\boldsymbol{x})).$$

Since Clifford group acts on the scalar compoonent identically, we get.

$$\rho(\boldsymbol{w})(\boldsymbol{x} \otimes p_{\boldsymbol{\theta}}(\boldsymbol{x}))^{(0)} = (\boldsymbol{x} \otimes p_{\boldsymbol{\theta}}(\boldsymbol{x}))^{(0)}.$$

Similar argument holds for $\|p_{\boldsymbol{\theta}}(\boldsymbol{x})^{(0)}\|$, and this completes the proof. □

*Proof of Corollary 3.2.*

**Sparse case**. Since the scoring function is invariant to the Clifford group action, the pooled nodes are same across the inputs acted by any $\boldsymbol{\omega} \in \Gamma(\mathbb{R}^n, q)$. The pooled nodes are sampled from the nodes in a graph, the features on the sampled nodes are equivariant to $\Gamma(\mathbb{R}^n, q)$. Unpooled nodes on the other hand are equipped with invariant features, such as zero multi-vectors or mean of the multi-vectors in the Clifford space, the unpooling operation is also equivariant to $\Gamma(\mathbb{R}^n, q)$-action.

**Dense case**. Note that the assignment matrix $S$ is invariant to $\Gamma(\mathbb{R}^n, q)$-action, since the multi-blade projection is $\Gamma(\mathbb{R}^n, q)$-invariant. By Theorem 3.2 of Ruhe et al. (2023), linear transform is $\Gamma(\mathbb{R}^n, q)$-equivariant, and therefore Equation 4 is $\Gamma(\mathbb{R}^n, q)$-equivariant. On the other hand, the adjacent matrix $A^{(h+1)}$ in equation 5 is invariant to $\Gamma(\mathbb{R}^n, q)$-action, since $S$ is $\Gamma(\mathbb{R}^n, q)$-invariant. Therefore, the dense pooling method is $\Gamma(\mathbb{R}^n, q)$-equivariant. Similar argument applies to the equation 6, therefore the dense unpooling is also $\Gamma(\mathbb{R}^n, q)$-equivariant. □

## D   EXPERIMENTAL SETTINGS

**Multi N-Body dataset**. The Multi N-Body dataset is based on (Kipf et al., 2018; Han et al., 2022) and it consists of $N$ charged particles $\{\boldsymbol{x}_i, \boldsymbol{v}_i, \boldsymbol{c}_i\}_{i=1}^N$ that belongs to $M$ disjoint complex systems $\{\mathcal{S}_j\}_{j=1}^M$, where $\boldsymbol{x}_i, \boldsymbol{v}_i, \boldsymbol{c}_i$ are particles' position, velocity and change. Withing each system $\mathcal{S}_j$ the particples are rigidly connected in pairs (bonds), triangles and tetrahedrons. The dynamics of the $M$ systems is driven by electromagnetic forces between pairs of particles. The task consists of predicting the final position of the all the particles after $T = 1500$ steps, given the initial positions and velocities. $J$ independent systems are samples, each composed of $M$ independent systems, with the number of particples sampled from uniform distribution with mean $N/M$. This system is called $(M, N/M, J)$. Withing each system $\mathcal{S}_j$ the particples are rigidly connected in pairs (bonds), triangles and tetrahedrons.

As graph topologies over which we perform message passing with HCGE-MPNN, we use the same graph topologies $A_{global}$ and $A_{local}$ as adopted in Han et al. (2022). In this experiment, we model $p_{\boldsymbol{\theta}}$ with CGE-MPNN and use $A_{local}$ as an input of $p_{\boldsymbol{\theta}}$ to compute the scoring function of **CE-Pool**$^{(h)}$. $A_{global}$ is used to update node features by CGE-MPNNs composing CE-Encode and CE-Decode. Both $A_{local}$ and $A_{global}$ are pooled based on the scores computed over $A_{local}$, and pooled $A_{local}$ and $A_{global}$ are used in the next round of pooling process.

Throughout this experiment, we set 2-layers and 14-channels for hidden layers of CGE-MPNN in our model. Pooling depth is set to be 1 and the ratio of pooled nodes (for sparse pooling) is 0.5. For dense pooling, the number of cluster is set to be same as the number of system in each of the experiments. For the N-body simulation system, we compared with Linear Prediction, EGNN (Satorras et al., 2021), EGHN (Han et al., 2022), following the implementation of Han et al. (2022). The accuracy of the prediction is measured in mean-squared error (MSE).

**Motion capture dataset**. The CMU Motion Capture Database CMU (2003) is a large collection of motion capture recordings for various tasks (such as walking, running, and dancing) performed by human subjects. We here focus on recorded walking motion data of a single subject (subject ♯ 35). The data is in the form of 31 3D trajectories, each tracking a single joint of the human body. For the training dataset, we adopt the same splitting strategy as Han et al. (2022), with 1100 frame pairs [1] for training, 600 for validation, and another 600 for tests. The interval between each pair is 30 frames in both scenarios. In this task nodes are the joints and edges are defined as the neighbor of nodes. We again use the same graph topologies $A_{global}$ and $A_{local}$ as adopted in Han et al. (2022).

For the number of clusters $K$ for dense pooling, we empirically find that $K = 4$ yields promising results. We set 2-layers and 14-channels for hidden layers of CGE-MPNN in our proposed method. The pooling ratio for the sparse pooling is 0.5. In this experiment, we compare HCGE-MPNN with Euclidean group equivariant models such as EGNN (Satorras et al., 2021) and EGHN (Han et al., 2022) as well as CGE-MPNN. The prediction accuracy is measured in mean-squared error (MSE).

## E   RESULTS OF DENSE POOLING

**Multi N-body dataset**. Table 3 is the result of N-body dataset experiments. Our sparse-pooling mode performs better or competitive to CGE-MPNN. However, the dense-pooling mode performs worse even than CGE-MPNN. We observed that our model with the both modes suffers from severe overfitting during training even with small hyperparamters such as message-passing layers or the channel dimension of CGE-MPNN. This could happen since the clifford representation of the

---

[1] Han et al. (2022) uses 200 frame pairs for the training data, but the reported scores for the walk experiment in the paper was not reproduced. We instead use 1100 frame pairs and train baseline models on them.

canonical vectors retains abundant information compared to the original vectors, which has huge impact on overfitting when the training data is small – we followed the same setting as that in Han et al. (2022), in which the training samples are 1,500 and the samples of validation and test datasets are 2,000. We plan to increase the number of training samples to see if the tendency is observed in this case.

**Motion capture dataset**. Table 2 is the result of motion capture dataset. Both of the pooling modes outperform all the baselines and the dense-pooling mode achieves the best performance among the others. In general, the dense-pooling mode pools nodes to create small graphs with smaller diameters to facilitate propagation of graph information while keeping the number of message-passing layers small. Therefore, the result indicates the need of dense-pooling method, since the graph topology of the data is sparse and its diameter is about 15, which require the flat message passing methods to have deep message passing layers to capture long-range dependency involving taking risks of over-smoothing or over-squashing. Although the sparse-pooling method also outperforms baseline methods, this mode essentially does not facilitate the message propagation in a manner of dense-pooling since it essentially drops nodes and does not increase the connectivity of the pooled graph.

Table 3: **Prediction error ($\times 10^{-2}$) of dense and sparse pooling modes on Multi N-Body dataset**. The "Multiple System" contains $J = 5$ different systems. For each column, $(M, N/M)$ indicates that each system contains $M$ complexes of average size $N/M$. Results averaged across 3 runs.

| | Single System ($J = 1$) | | Multiple Systems ($J = 5$) | |
|---|---|---|---|---|
| | (3,3) | (5,5) | (3,3) | (5,5) |
| EGNN (Satorras et al., 2021) | 12.69 (0.19) | 15.37 (0.13) | 13.33 (0.12) | 15.48 (0.16) |
| EGHN (Han et al., 2022) | 11.58 (0.10) | 14.42 (0.80) | 12.80 (0.56) | 14.85 (0.30) |
| CGE-MPNN (Ruhe et al., 2023) | 9.44 (0.41) | **8.18** (0.14) | 9.60 (0.21) | **7.18** (0.36) |
| HCGE-MPNN (dense) | 11.63 (1.45) | 10.95 (1.39) | 11.79 (1.28) | 9.34 (0.64) |
| HCGE-MPNN (sparse) | **8.75** (0.24) | 8.21 (0.13) | **9.14** (0.40) | 7.29 (0.61) |

## F  ABLATION STUDY

We perform the ablation study of HCGE-MPNN with the motiona capture dataset. We conducted the study for each of the dense and sparse modes and report the scores in Table 4. For our dense pooling method, we change the number of clusters to 3 and 7, both of which yield worse performance. We also modify the hidden channels from 14 to 8, and we observed that HCGE-MPNN with 14 hidden channels performs better. Notably if we omit the connectivity loss, in whcih the topology of poolded graphs are regularized to ensure being sparse, its performance degrades. This shows the importance of having the connectivity loss in the training objective. We also investigate the necessity of having sparse pooling for this experiment. We modify the hidden channel from 14 to 8 and pooling ratio from 0.5 to 0.2 and 1.0, all of which degrades its performance. Setting pooling ratio to be 1.0 is equivalent to CGE-MPNN model with the skip connection. We conclude this section by noting that we did not perform experiments with larger hidden-channels and larger hidden layers to keep the maximum computational budget same across all the ablation models.

Table 4: **Ablation study on motion-capture dataset**. Prediction error is set to be MSE ($\times 10^{-2}$) and results are averaged across 3 runs.

| HCGE-MPNN (Dense) | Walk (#35) |
|---|---|
| hidden channels = 8 | 7.2 |
| hidden channels = 14 | **6.0** |
| clusters = 3 | 7.1 |
| clusters = 7 | 6.5 |
| w/o hierarchy | 7.7 |
| w/o Connectivity loss | 6.7 |

| HCGE-MPNN (Sparse) | Walk (#35) |
|---|---|
| hidden channels = 8 | 7.5 |
| hidden channels = 14 | **6.3** |
| pooling ratio = 0.2 | 8.5 |
| pooling ratio = 1.0 | 7.2 |

## G  RELATED WORKS

### HIERARCHICAL GRAPH NEURAL NETWORKS (POOLING AND UNPOOLING)

Hierarchy is common in various domains and it is typically employed in the concept of information coarsening, where multi-resolution representations of information are obtained at different levels of abstraction along the hierarchy. Coarsening provides simple, yet effective methods to extract features, and the idea is key in image-processing Fukushima (1980); LeCun et al. (1989) and has recently played a vital role in unstructured-data processing Gao & Ji (2019); Cangea et al. (2018); Lee et al. (2019); Ying et al. (2018); Bianchi et al. (2020); Khasahmadi et al. (2020); Yuan & Ji (2020). These studies focus mainly on generic data, and more importantly, they do not respect the symmetry of geometric data stemmed from physical systems.

Graph Neural Networks (GNNs) have achieved a substantial improvement over various tasks, from node-level tasks such as node classification to graph-level tasks such as graph regression and classification. In order to obtain an effective graph representation, many designs of graph hierarchical pooling, an operation to iteratively coarsen graphs into smaller size, have been proposed. These methods typically involve learnable score functions to compute likelihood of nodes to be pooled. Such formulation includes *sparse* method (Gao & Ji, 2019; Cangea et al., 2018; Lee et al., 2019; Knyazev et al., 2019; Zhang et al., 2020; Ranjan et al., 2019; Ma et al., 2020), which computes a score for each node and reduces the graph by keeping only a fraction of top scoring ones and dropping the rest, and *dense* method (Ying et al., 2018; Bianchi et al., 2020; Khasahmadi et al., 2020; Yuan & Ji, 2020; Tsitsulin et al., 2023) that computes for each node a soft-assignment to a fixed number of clusters. Although the methods can effectively obtain representation of graphs, the methods do not respect symmetry of physical systems, which limits their generalization on real-world geometric data such as ones in the context of multi-scale modelling. One important exception is the work of Han et al. (2022), that is formulated in an equivariant encoder-decoder form with skip-connections. Our method is also aligned with the encoder-decoder form, but instead respects Clifford group action, that involves higher-order action to geometrical data.

## H  VISUALIZATION

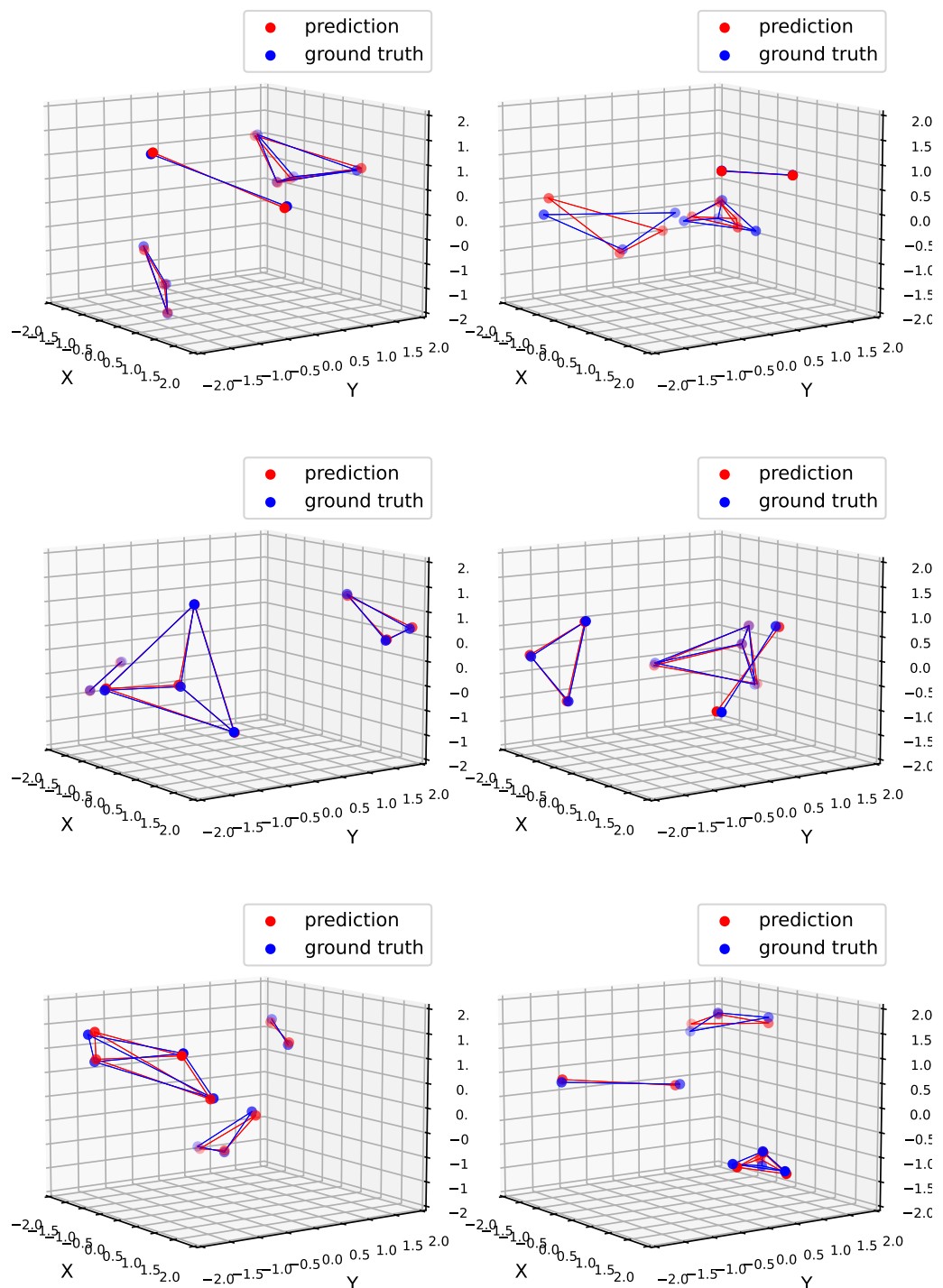

Figure 2: **Visualization of the output of HCGE-MPNN and ground-truth for Multi N-Body dataset.** Figures represent the 3d coordinates of the particles composing (3, 3, 1)-system. Edges correspond to the frame of each sub-system and each sub-system is fully connected.

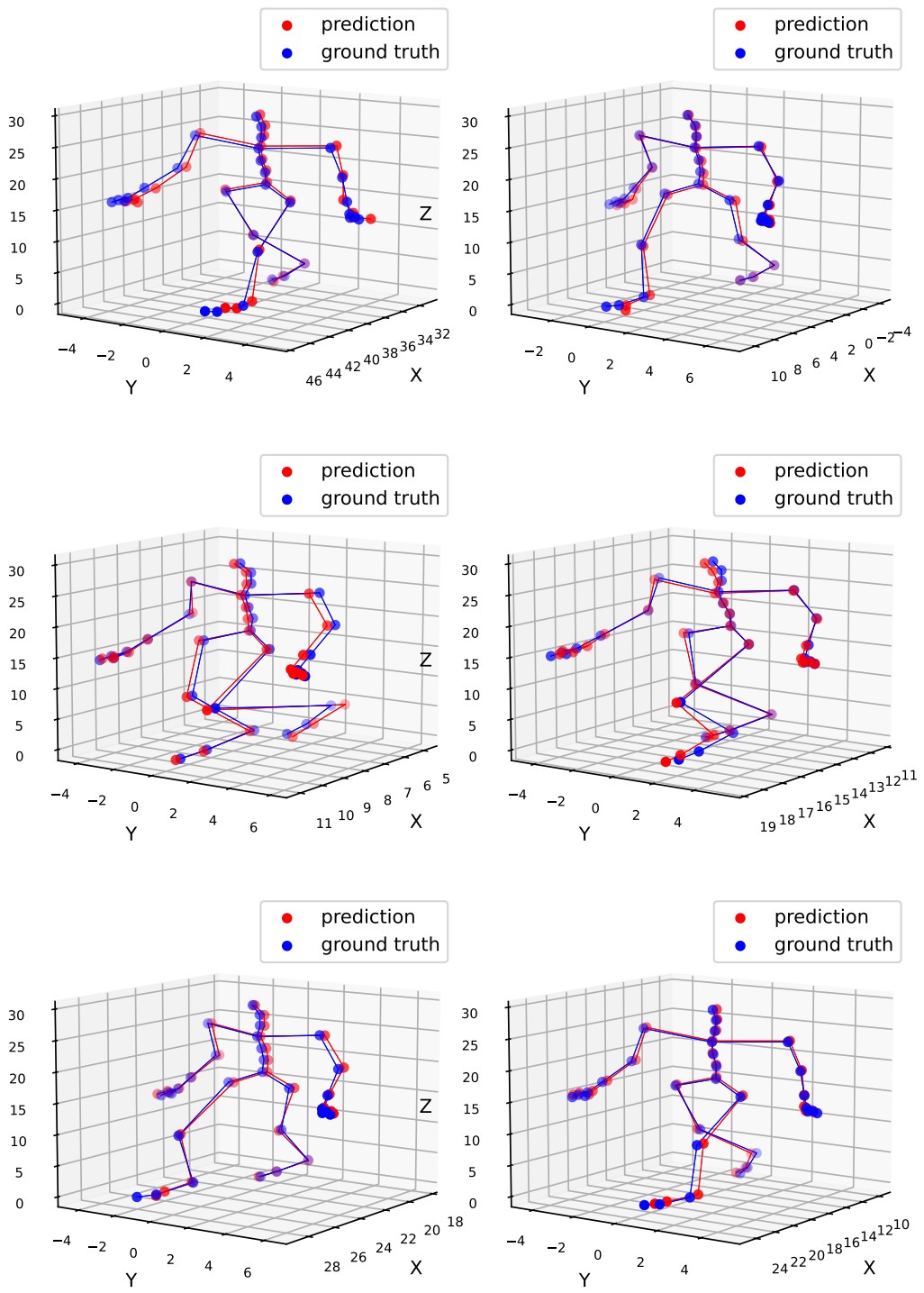

Figure 3: **Visualization of the output of HCGE-MPNN and ground-truth for motion capture dataset.** Figures represent the 3d coordinates of the joints representing body.

