# OpenReview forum: "Hierarchy-based Clifford Group Equivariant Message Passing Neural  Networks"
_ICLR.cc/2024/Workshop/AI4DiffEqtnsInSci — AI4DiffEqtnsInSci @ ICLR 2024 Poster_

### Official Review · Reviewer_X3so · 2024-02-26
**Great contribution about the use of U-Net.**

**Rating:** 8
**Confidence:** 3

**Review:**

This is an outstanding work. It is very well written and the story line flows very smoothly. Even for a non expert in the topic, I could understand the motivation, the flowchart of the methodology and the impressive results reached. It is also an important contribution for the modeling of physical systems using Neural Networks.

Pros:
* The description of the data used and the methods contains enough details to have a clear understanding of what they did.
* They clearly state the limitations of their results and provide insights of why their methods don't reach sometimes the best performance.
* They provide an outlook of next steps to improve their results.

Cons:
* There is an overfitting problem that might make their results not so appealing.
* I felt that the authors missed to emphasize the novelty of their contribution.

---

### Meta-Review · Area_Chair_DJro · 2024-03-03

**Recommendation:** Accept (Poster)

**Metareview:**

This paper introduces a Hierarchy-based Clifford Group Equivariant Message Passing Neural Network (HCGE-MPNN), a Clifford group equivariant U-Net with skip-connection. The paper is well written and a useful contribution to the field and should be accepted for a poster session.

---

### Decision · Program_Chairs · 2024-03-03

Accept (Poster)